# Preparedness for Disaster Response: An Assessment of Northeast Romanian Emergency Healthcare Workers

**DOI:** 10.3390/healthcare13182257

**Published:** 2025-09-09

**Authors:** Alexandra Haută, Radu-Alexandru Iacobescu, Paul Lucian Nedelea, Mihaela Corlade-Andrei, Tudor Ovidiu Popa, Carmen Diana Cimpoeșu

**Affiliations:** 1Department of Emergency Medicine, Surgery II, University of Medicine and Pharmacy “Grigore T. Popa”, 700115 Iași, Romania; h_alecsandra@yahoo.com (A.H.); paul.nedelea@yahoo.com (P.L.N.); corladeandrei.mihaela@yahoo.com (M.C.-A.); tudor.popa@umfiasi.ro (T.O.P.); dcimpoiesu@yahoo.com (C.D.C.); 2Emergency Care Department, “Sf. Spiridon” County University Emergency Hospital, 700111 Iași, Romania; 3Department of Nursing, Internal Medicine II, University of Medicine and Pharmacy “Grigore T. Popa”, 700115 Iași, Romania; 4Thoracic Surgery Department, Respiratory Disease Hospital, 700115 Iași, Romania

**Keywords:** disaster preparedness, emergency care, emergency personnel, disaster training, trauma training, readiness to practice

## Abstract

Background: Disasters, although predictable, often occur unexpectedly, and efforts must be directed towards reducing their impact. Emergency healthcare workers, key players in disaster response, should maintain a high level of preparedness to act in catastrophic situations. Data on knowledge, attitude, and disaster preparedness among emergency healthcare workers is scarce, particularly for developed countries in Europe. This study aimed to measure the perceived preparedness of various health practitioners in emergency care in Iași county (Romania) and identify factors that influence it. Materials and methods: A self-assessment web-based questionnaire was developed to measure knowledge (K), attitude (A), and preparedness (P). Nonparametric tests compared measurements between demographic groups. Spearman correlation, linear univariate, and multivariate regression models were used to test the effect of perceived knowledge, attitude, and other work-related factors (such as experience, training, and leadership) on disaster preparedness. Results: 211 valid entries were recorded (114 female and 97 male), of which 33.6% were doctors, 25.1% were nurses, and 23.7% were paramedics. There were differences in exposure to training across health professions for disasters and trauma management (*p* = 0.03 and *p* = 0.009). The sample’s overall scores for the three primary domains assessed were moderate. Univariate analyses identified a significant effect of knowledge and attitude on preparedness (B = 0.9, 95% CI: 0.79–1.01, *p* < 0.001, and B = 0.81, 95% CI: 0.66–0.97, *p* < 0.001, respectively), which was maintained in multivariate regression. Workplace factors (disaster plans and institutional collaboration), along with experience in disaster management and emergency care, were determinants of preparedness, while the effect of training was insignificant. Conclusions: Most healthcare workers displayed moderate preparedness for disasters, while exposure to training and practice was found to be inadequate. Focus should be placed on identifying barriers and enhancing training delivery, strengthening institutional involvement in staff preparedness, and improving inter-professional collaborations. Adequate training methods must be developed and validated in further studies.

## 1. Introduction

Disasters, related to natural phenomena or human activity, have an upward incidence worldwide, potentially causing disruption at the community level, overwhelming local infrastructure capacity, and resulting in significant material or life loss [1,2]. Since some are predictable and preventable (although many remain unexpected), the United Nations focused on mitigating the impact these have on communities and developed the SENDAI framework, which emphasizes understanding risks, strengthening governance, increasing resilience, and enhancing disaster preparedness, particularly among emergency first responders and healthcare workers (HCWs) [3].

Emergency care workers are the front line of any emergency disaster response, and their involvement in rescue efforts is essential to disaster management success [4,5]. Their preparedness has been shown to depend on factors such as training, previous experience, availability of institutional disaster plans, and individual perception of self-efficacy [6,7]. Data, mainly from the Middle East and Asia, indicate that preparedness among HCWs is variable, which emphasizes the need for its assessment and improvement, particularly in disaster-prone regions [8].

In Europe, the assessment of HCWs’ preparedness remains critical, and is so far limited, specifically considering recent events, like the floods in Spain and Romania, which underscore how global unpreparedness leads to health and economic fallout [9,10]. This is most true for the northeast region of Romania, a region at the crossroads of repeated natural calamities and geopolitical tension (given the proximity to ongoing military conflict), which makes it particularly vulnerable to disasters. Romania, a European Union (EU) member state, has sought to strengthen its emergency healthcare system and has a few well-trained teams ready to participate nationally and internationally through the WHO EMT (World Health Organization Emergency Medical Teams) and the EU Civil Protection initiative rescue efforts [11,12]. However, a comprehensive global view of disaster preparedness is lacking, and community mistrust and low engagement remain challenges in this country’s disaster preparedness; therefore, a further assessment of HCWs’ preparedness is necessary [13,14].

In this study, we aimed to measure self-reported knowledge, attitude, and preparedness regarding disasters among emergency healthcare personnel in Iași County, the key region for emergency healthcare in the northeast part of the country. Another goal of this study was to identify factors that influenced preparedness and to pinpoint areas for improvement.

## 2. Materials and Methods

### 2.1. Design

A survey-based, cross-sectional, quantitative study on emergency healthcare workers from Iași County, Romania, was conducted in February 2025.

### 2.2. Study Participants and Sampling Procedure

A convenience sampling technique was used for the purpose of this study due to logistical considerations, such as accessibility, to cover as large and representative a sample of HCWs within the county emergency services as possible. Participants were approached to respond to the survey via email and institutional emergency department WhatsApp contact groups, which were commonly used communication methods within these professional networks. The County University Emergency Hospital “Sf. Spiridon” in Iași serves as a regional coordination center for emergency care in the northeast region of the country and has provided professional contact information for active professionals within the network. This network includes six institutions and emergency care departments within the county. A web-based survey was distributed to any available participants and completed through an online platform (Google Forms). This approach allowed timely access to willing participants but limited participation to professionals actively engaged in these communication networks.

A wide range of professionals were included in this study in order to capture a comprehensive perspective on preparedness, given their different roles in emergency care. All HCWs (doctors, nurses, paramedics, and auxiliary personnel) were eligible to participate if they actively provided care to adults in the emergency department or prehospital emergency services, and their participation was voluntary. Professionals providing care in pediatric emergency departments were not included because their curricula and daily practices differed significantly and were not the focus of this study. A completion time frame from the first of February until the 14th of the month was set for early responders. After this time interval, emails and messages were sent to reinvite participants to complete the survey, and all responses following the 15th of February were considered late responders. This was performed to account for any non-response bias that may have occurred [15]. Entries were removed if the participants did not consent to participate in the study and if data in the survey were missing.

The sample size needed for statistical significance was calculated using Gpower (v. 3.1) [Computer software]. Based on a significance level of 0.05, a medium effect size (0.15), and a Power (1-β) of 0.8, a minimum required sample size of 143 participants was determined.

### 2.3. Assessment Instrument and Validity

A self-assessment questionnaire was developed based on findings from the literature on similar validated instruments [16,17,18]. Questions in the assessment were adapted to suit the population studied and the aims of the research. The primary measurement in the survey was participants’ self-reported knowledge (K), attitude (A), and preparedness (P) in the event of a disaster, and the questions used for their assessment are available in Appendix A (Table A1). The demographic section of the survey assessed factors that could influence perceived preparedness, such as gender, age, years of experience in emergency care, exposure to education in trauma and disasters, experience in disaster management, and work environment factors. A five-point Likert scale was used to evaluate the three domains (responses ranged from Strongly agree = 5, Agree = 4, Neither agree nor disagree = 3, Disagree = 2, and Strongly disagree = 1). Knowledge was assessed with 11 questions, yielding scores from 11 to 55. The cutoff values for low, moderate, and high scores were considered to be 22, between 22 and 44, and above 44, corresponding to <25th percentile, 25th–75th percentile, and >75th percentile, respectively. Similarly, 13 survey items assessed attitude, with total scores ranging from 13 to 65. Values set as cutoff values for low, moderate, and high were 41, between 41 and 57, and above 57. Lastly, 15 items evaluated preparedness, with total scores from 15 to 75. Scores less than 34 indicated low preparedness, 34 to 60 indicated moderate, and scores above 60 indicated high preparedness.

The survey tool was piloted on 85 emergency healthcare practitioners to adjust question appropriateness. Internal validity for each item was assessed using Cronbach’s alpha and indicated good results for all scores (Knowledge-0.86, Attitude-0.84, and Preparedness-0.88, respectively). Minor adjustments were made. Further factor analysis was performed using the Kaiser–Meyer–Olkin method, and significance was assessed using Bartlett’s test of sphericity. Values for K, A, and P were 0.85 (*p* < 0.001), 0.80 (*p* < 0.001), and 0.83 (*p* < 0.001), respectively, which meant that the use of answers as a single score for each item was appropriate.

### 2.4. Statistical Analyses

Statistical analyses were performed using SPSS (v. 23, IBM Corp., New York, NY, USA) [Computer software]. The normality of distribution was assessed using the Shapiro–Wilk test. Because the normality of the distribution condition was not met, the Mann–Whitney U Test and Kruskal–Wallis Test were performed to compare continuous data between groups, with median and interquartile ranges (Q1–Q3) being reported. Post hoc analyses using Dunn–Bonferroni correction were further performed to assess between-group comparisons. Categorical data comparison was performed using the χ^2^ test, and data were reported as total count (n) and frequencies (%). Correlations between continuous data and primary measurements were performed using Spearman correlation. Univariate linear regression was used to assess the effect of age, years of experience, knowledge, and attitude on preparedness scores. A further multivariate linear regression model was developed to assess demographic factors influencing the primary outcome variable (P scores). Covariates used in the multivariate model were chosen based on exploratory findings and included gender, profession, level of education, years of experience in the emergency healthcare service, trauma courses, disaster management courses, experience in disaster management, involvement in disaster prevention, involvement in disaster education, presence of disaster plan at an institutional level, department involvement in disaster education, between-institution collaborations for disaster management, knowledge, and attitude scores. Multicollinearity within the models was assessed using VIF (Variance Inflation Factor) and by assessing tolerance. Further subgroup analyses were performed to assess how the identified factors influenced each professional group’s preparedness (doctors, nurses, and paramedics, respectively). This involved conducting separate regression analyses, as described above, for each professional category. The significance level was set at α = 0.05 for all statistical tests performed.

### 2.5. Ethics

Ethical review board approval for the survey was obtained from the County Emergency Care Hospital “Sf. Spiridon”, Iași (approval number 101 from 14 November 2024), and further approval was sought from the University of Medicine and Pharmacy “Grigore T Popa,” Iași (approval number 521 from 25 January 2025). Participants consented to participate in the study before completing the survey. All research was performed in accordance with the 2024 revised Declaration of Helsinki.

## 3. Results

Out of the 685 active professionals within the county’s emergency care infrastructure, 444 were invited to participate in the survey. Of those, 214 healthcare workers responded (response rate of 48.2%), with 3 providing incomplete answers and being excluded. The final sample consisted of 211 participants, primarily emergency department resident doctors, nurses, and paramedics. Statistically significant differences in gender distribution (*p* < 0.001) among professionals were observed, as most nurses, residents, specialists, and senior doctors were female (73.58%, 67.61%, 58.33%, and 71.43% of each group, respectively), while paramedics were predominantly male (92%). The respondents were between 22 and 67 years old, with a median of 33 (IQR: 28–42), with a statistically significant difference between genders as female participants were slightly older (a median of 33 for males vs. 35 for females, *p* = 0.03). There were no significant differences between early and late responders except for age (median 32 for early and 35 for late, *p* = 0.02).

Exposure to training varied across the sample. A significant difference was found between emergency care professions and trauma care training exposure (*p* = 0.009). The majority of residents did not participate in any trauma care courses (57.75%), whereas most specialists, senior doctors, nurses, and paramedics did (83.33%, 78.57%, 66.04%, and 62%, respectively). Differences were also observed in exposure to practice (*p* = 0.03), with 63.38% of residents and 56.6% of nurses reporting they had never been involved in a disaster care situation, while most specialists, senior doctors, and paramedics had (66.67%, 78.57%, and 56%). In total, 134 (63.51%) participants received some form of disaster management training, with no differences across genders and professions. Table 1 summarizes sample characteristics and shows the association with knowledge, attitude, preparedness, and total KAP scores.

Knowledge scores were found to be moderate in 69.67% of responses and high in 27.49%, with an overall median of 54 (IQR: 49–59). These were higher among the male group (42, IQR: 37–46 vs. 39, IQR: 32–45, *p* = 0.005), and a significant difference among professions was found (*p* < 0.001). Post hoc analysis revealed lower knowledge scores of residents compared with specialists, doctors, nurses, and paramedics (*p* value 0.002, 0.05, and <0.001, respectively). Median values and IQRs are shown in Table 1 for comparison. Increased knowledge scores were observed among participants with adequate exposure to education, like trauma courses (*p* < 0.001) and disaster courses (*p* < 0.001), and among participants actively involved in disaster education (*p* < 0.001). Exposure to hands-on experience in disaster management also significantly increased participants’ knowledge scores (*p* < 0.001), while years of experience in the emergency service weakly correlated with knowledge scores (Rs 0.35, *p* < 0.001). Regarding disaster training, post hoc analyses revealed significance only between education forms and no education, without differences among teaching methods. Differences in knowledge scores and time since the last course (*p* < 0.001) were also due to the comparison with the group that never received any training. Higher scores were also observed if participants declared adequate institutional involvement (like teaching and disaster plans set in place, *p* = 0.002 and *p* < 0.001, respectively).

Attitude scores were moderate in 65.4% and high in 30.33% of cases, with no statistically significant difference among professions and gender. Similarly to knowledge scores, attitude scores were also influenced by education, exposure, and the workplace environment. Of interest, higher attitude scores were reported by participants whose workplace was run by an emergency care specialist (*p* = 0.009).

The median score for preparedness was 50 (IQR: 43.5–55.5). The large majority showed moderate preparedness (82.94%), while a small proportion presented high scores (11.85%). There was a significant difference between genders, with males showing higher scores (median 53 for males vs. 47 for females, *p* < 0.001). Among professions, significantly lower scores were observed for residents compared with paramedics and emergency specialists (*p* < 0.001 and *p* = 0.015) and for nurses compared with paramedics (*p* = 0.036). If there was exposure to education, practice, and adequate institutional preparedness, statistically significant higher scores were again observed.

We found that there was a strong positive correlation between knowledge and preparedness (Rs 0.78, *p* < 0.001) and between attitude and preparedness (Rs 0.55, *p* < 0.001) (Figure 1). Univariate linear regression (Table 2) shows that knowledge and attitude have significant effects on self-reported preparedness (B = 0.9, 95% CI: 0.79–1.01, *p* < 0.001 for knowledge and B = 0.81, 95% CI: 0.66–0.97, *p* < 0.001 for attitude). Interestingly, multivariate linear regression shows that while the effect of knowledge and attitude on preparedness was maintained, and there was an effect of experience in emergency care and disaster management, the training effect on preparedness was insignificant (Table 3). Other important factors identified were the existence of disaster plans at the level of the institution and collaboration between institutions for disaster management. No multicollinearity was detected among assessed variables. For each specific profession, the correlation between knowledge, attitude, and preparedness remained true (Appendix B Table A2). However, the other factors influencing preparedness varied, as reported in Appendix B Table A3. According to these data, previous disaster experience and the existence of institutional collaborations were significant factors for doctors, while for nurses, readily available disaster plans were an important determinant. For paramedics, in multivariable linear regression, only knowledge and attitude determined their level of self-assessed preparedness for disaster.

## 4. Discussion

Preparedness is a vital component of the disaster cycle framework, alongside mitigation, response, and recovery, and it is essential for building disaster resilience [19]. Adequate disaster management depends on experienced, well-prepared, and ready-to-act emergency healthcare personnel. So far, data about practitioners’ preparedness for disasters and factors that influence it remain limited, and its assessment within developed countries from the EU is critical. Our survey from Iași County, Romania, sought to assess the preparedness of emergency healthcare workers from a relevant disaster-prone region and identify factors that influenced preparedness in practice. The findings show that preparedness among HCWs is moderate and that self-reported knowledge and attitude toward disasters are relevant determinants of preparedness. This is in line with other recently reported results in developed countries like the UAE (United Arab Emirates) and China, as well as in developing countries like Pakistan and Yemen [16,18,20,21]. Additionally, we identified that training, experience, and workplace factors (such as disaster plans, leadership, involvement in education, and institutional collaborations) varied within our sample, played an important role in overall preparedness, and could serve as potential targets for interventions to improve preparedness.

Perhaps one of the most critical vulnerabilities in disaster preparedness identified in our study is the lack of training uniformity across professions. This can, among others, be explained by factors such as differences in curriculum needs for each profession, accessibility to training programs, and affordability. Disaster training and curricula contents are facing continuous adjustments as the role of education is occupying its rightful place in disaster risk reduction strategies [22,23]. An updated curriculum is now available at the postgraduate level, and a standardized competency model is available for European Union member states to guide training and ensure consistency of practice [24,25]. However, the uptake and effect of training on increasing preparedness are unknown. Several other countries have also observed inadequate training among nurses- and doctors-in-training [20,26,27,28]. The importance of training cannot be overstated. In our sample, for instance, participants who underwent training had significantly increased self-reported knowledge, attitude, and preparedness scores. Still, the results failed to provide evidence of superiority for a specific training method. Furthermore, the impact of training in our multivariate linear regression analysis was lost. This may be due to the variation in training opportunities accessed and the differing intervals between refresher courses for different professionals in our study. Further research is needed to determine how training influences preparedness over time and to identify the most effective training method for disaster preparedness. Nonetheless, our data supports implementing regular, standardized disaster training for all healthcare workers, customized to meet their specific needs.

Besides individual education, our survey also identified the importance of workplace factors, particularly for hospital practitioners. Key points where institutions could increase disaster preparedness of HCWs, identified in the literature, were the existence of disaster plans, institutional involvement in education, communication, and collaboration with external partners [29]. Our subgroup analyses also highlight the importance of these for practitioners working within this environment. In addition, our data shows the importance of leadership in increasing attitudes toward disasters, as practitioners with emergency care specialist guidance presented higher attitude scores. The important role of leadership has been shown in training for trauma management [30]. It is now emerging as a significant factor in determining preparedness, particularly for nursing practitioners [31]. Maintaining adequately prepared leadership at an institutional level could be a key factor in increasing resilience when disaster training cannot cover all healthcare staff. Thus, in order to increase the overall preparedness of hospital HCWs, institutions should establish clear disaster plans and foster adequate leadership.

Another relevant factor that has emerged is the role of previous experience and workplace involvement in sustaining adequate levels of preparedness. An assessment of healthcare workers from Canada who participated in international disaster management efforts highlighted the importance of these experiences in improving efficiency and team capacity to cope [32]. In our assessment, exposure to practice was a key factor in determining preparedness. Professionals who often take part in disaster management efforts, such as paramedics, displayed consistently high levels of preparedness. Hands-on experience for residents could be improved by including these doctors-in-training in management efforts of events with potential mass casualties, such as mass gatherings. Mass gathering events can be a potential training ground for disaster management and may serve an additional purpose for some healthcare professionals [31,33]. Providing opportunities for on-the-field experience in already established teams could also be appropriate.

Finally, our analyses identified significant gender disparities in knowledge and preparedness, with males exhibiting higher scores. Gender was a significant factor in our multivariate linear regression overall; however, this significance was lost in subgroup analyses. The significant difference in gender distribution among professions and in exposure to practice in disaster management among professions can account for the observed difference and association to preparedness in our sample, as most males were paramedics who, as a profession, had significantly higher exposure to practice in disaster management compared with other professions. In studies where gender was equally distributed among health professions, no such differences were observed [16]. For comparison, in studies with disproportionate gender distribution among professions investigated, similar disparities were found [17,18]. Further efforts should concentrate on closing the gender gaps in training and providing equal practice opportunities for female HCWs to further strengthen systemic disaster resilience.

### Study Limitations

This study assessed knowledge, attitude, and perceived preparedness for disaster situations within a representative and diverse sample of emergency care providers. However, despite sampling nearly a third of the entire target population, the survey’s response rate was moderate, and some opinions may have been underrepresented. Furthermore, the convenience sampling method may introduce sampling bias, and thus, there might be differences between the sample and the population investigated. The data reflect only a brief time period due to the cross-sectional design. Therefore, no conclusions can be drawn about how education and training impact over time. Information about training exposure collected in the survey was also limited. It does not allow for the duration of exposure, type of training, or teaching methods used to be compared; thus, the conclusions on training impact must be considered carefully. The measurements reported are based on a self-reported questionnaire and might be influenced by social desirability and cognitive bias (the Dunning–Kruger effect) [34]. Unmeasured potential confounders, such as intrinsic motivations for practice or psychological resilience, might influence these results. Finally, an insufficient sample size in some groups could influence the relevance of some of the post hoc analyses, as it could reduce the reliability of the detected effects.

## 5. Conclusions

In our survey, most emergency healthcare professionals displayed moderate knowledge, attitude, and preparedness regarding disasters, mainly due to uneven exposure to training and practice. We found that addressing gender gaps in training opportunities, ensuring tailored training according to professional needs, and increasing institutional involvement through leadership, training, and interinstitutional collaboration are essential steps in improving preparedness and increasing disaster resilience in the assessed community. Further research should determine the specific education needs and barriers in education delivery for each profession, provide evidence of the best training approach, and determine its effect on preparedness over time.

## Figures and Tables

**Figure 1 healthcare-13-02257-f001:**
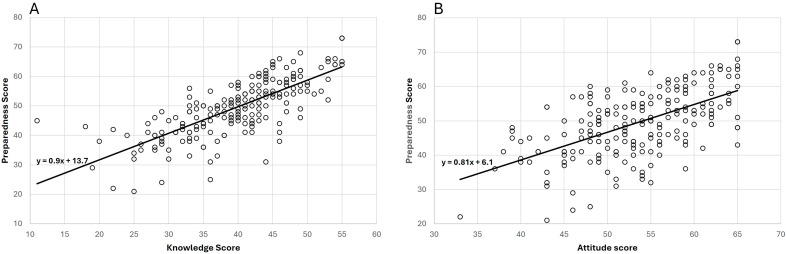
Correlation between knowledge and preparedness scores (**A**) and attitude and preparedness scores (**B**). Regression line and equation shown.

**Table 1 healthcare-13-02257-t001:** Demographic characteristics of healthcare workers and association with self-assessed Knowledge (K), Attitude (A), Preparedness (P), and total KAP scores.

Variables	Total Value	Knowledge (K)	*p*	Attitude (A)	*p*	Preparedness (P)	*p*	Total KAP	*p*
Gender			0.005		0.61		<0.001		<0.001
Male	97 (45.97%)	42 (37–46)		53 (49–59)		53 (47–58)		146 (134–161)	
Female	114 (54.03%)	39 (32–45)		54 (49–59)		47 (41–53)		139 (124–153)	
Age	33 (28–42)	0.28 *	<0.001	−0.07 *	0.31	0.06 *	0.35	0.11 *	0.12
Years of experience	6 (3–10.5)	0.35 *	<0.001	−0.05 *	0.43	0.18 *	0.01	0.18 *	0.009
Profession			<0.001		0.16		<0.001		0.002
Senior physician	14 (6.6%)	44 (33.48)		53.5 (45–57)		47.5 (41–55)		147.5 (119–161)	
Specialist	12 (5.7%)	45.5 (41–48)		57 (52–62.5)		58 (50–60)		161 (142–169.5)	
Resident	71 (33.6%)	36 (31–41.5)		54 (51–58)		46 (42–51)		137 (126–148.5)	
Nurse	53 (25.1%)	42 (35–46)		54 (48–59)		50 (41–56)		144 (124–160)	
Paramedic	50 (23.7%)	42.5 (39–48)		52 (48–59)		55 (48–60)		147.5 (137–166)	
Auxiliary staff	11 (5.2%)	40 (36.5–43.5)		48 (47–51.5)		47 (42.5–52)		137 (124.5–144.5)	
Level of education			<0.001		0.09		0.13		0.11
Post-secondary education	56 (26.54%)	40.5 (36.5–44.5)		50 (47.5–59)		50 (44–57)		142.5 (125–156.5)	
University education	131 (62.09%)	39 (32.5–44)		54 (50.5–58)		49 (43–55)		142 (126–155.5)	
Doctoral degree	24 (11.37%)	45 (41.5–49)		54 (48–61)		53.5 (47.5–59)		152 (134–168.5)	
Year of residency			0.09		0.76		0.18		0.2
First	8 (3.79%)	33 (30.5–41)		52.5 (49–56)		44 (41.5–50)		129 (122.5–146)	
Second	17 (8.06%)	33 (31–40)		56 (52–58)		48 (43–53)		142 (129–149)	
Third	17 (8.06%)	34 (29–37)		53 (52–58)		44 (40–49)		130 (121–142)	
Forth	16 (7.58%)	37 (34–41.5)		53.5 (48.5–59.5)		45.5 (42.5–50)		134.5 (126.5–150)	
Fifth	13 (6.16%)	41 (39–42)		55 (51–59)		50 (46–52)		146 (145–149)	
Trauma life support courses			<0.001		0.009		<0.001		<0.001
Yes	122 (57.82%)	42.5 (39–46)		55 (50–59)		52.5 (45–58)		148.5 (137–161)	
No	89 (42.18%)	37 (32–42)		52 (48–58)		46 (41–52)		134 (120–146)	
Disaster management education received			<0.001		<0.001		<0.001		<0.001
No education	77 (36.49%)	36 (32–41)		50 (47–54)		45 (39–50)		132 (118–143)	
Theoretic courses	23 (10.90%)	43 (37.5–45.5)		55 (51.5–57.5)		54 (49–57.5)		149 (142–158.5)	
Practice sessions	35 (16.59%)	44 (40–48)		58 (52–62)		55 (51.5–60)		157 (144–166.5)	
Hospital drills	45 (21.33%)	41 (35–46)		55 (49–60)		51 (44–57)		146 (130–163)	
Mixed methods	31 (14.69%)	42 (39–44.5)		55 (51.5–58)		51 (43.5–55)		147 (137.5–155.5)	
Disaster training organized by the department within the last 12 months			0.002		0.013		0.009		0.003
Yes	178 (84.36%)	41.5 (35–46)		54 (49–59)		51 (44–57)		145 (131–160)	
No	33 (15.64%)	37 (33–40)		50 (46–54)		47 (40–50)		133 (119–143)	
Time elapsed since the last disaster management course			<0.001		0.002		<0.001		<0.001
Never participated in any courses	60 (28.44%)	35 (30–40)		50.5 (46.5–54.5)		44 (38–50)		129 (117–144)	
More than 6 years ago	21 (9.95%)	42 (37–46)		50 (48–60)		50 (42–55)		143 (132–159)	
Within the past five years	38 (18.01%)	43 (39–47)		54.5 (49–59)		53 (47–57)		149 (137–162)	
Within the last two years	23 (10.90%)	43 (39–49)		57 (53–62)		57 (50–61)		158 (141.5–169)	
Within the last year	46 (21.80%)	41 (36–44)		54 (50–58)		50 (45–58)		146 (130–156)	
Within the last 6 months	23 (10.90%)	40 (37.5–44)		57 (51.6–61)		52 (46–56.5)		150 (138.5–162.5)	
Previous participation in disaster management			<0.001		0.002		<0.001		<0.001
Yes	102 (48.34%)	43 (38–47)		55 (49–60)		53.5 (46–59)		149 (137–164)	
No	109 (51.66%)	38 (33–42)		53 (48–57)		47 (41–53)		135 (123–149)	
Involvement in disaster prevention in the community			<0.001		0.026		<0.001		<0.001
Yes	87 (41.23%)	44 (39–49)		55 (50–60.5)		54 (47.5–59)		152 (136.5–166.5)	
No	124 (58.77%)	39 (33–43)		52.5 (48–58)		47 (41–52)		138 (124–149)	
Disaster plan at the institutional level			<0.001		0.047		<0.001		<0.001
Yes	193 (91.47%)	41 (35–46)		54 (49–59)		50 (40–57)		145 (130–160)	
No	18 (8.53%)	33.5 (27–39)		50.5 (46–54)		42.5 (34–50)		122.5 (113–136)	
Is the department leader an emergency care specialist?			0.35		0.009		0.79		0.19
Yes	164 (77.73%)	41 (35–45)		54 (49–59)		50 (44–55)		144 (129.5–158.5)	
No	47 (22.27%)	39 (33–46)		50 (45.5–58)		50 (42–57)		139 (120–152.5)	
Collaborations between institutions for disasters			0.06		0.98		0.044		0.12
Yes	202 (95.73%)	41 (35–45)		54 (49–59)		50 (44–56)		143.5 (128–158)	
No	9 (4.27%)	35 (33–37)		52 (51–59)		41 (36–49)		130 (120–140)	

* Rs-Spearman correlation coefficient shown.

**Table 2 healthcare-13-02257-t002:** Univariate linear regression for predictors of disaster preparedness.

Variable	Coefficient (B)	Standardized Coefficient (Beta)	95% CI for B	*p*
Age	0.02	0.01	−0.13 to 0.16	0.84
Years of experience	0.08	0.06	−0.12 to 0.27	0.42
Knowledge score	0.9	0.75	0.79 to 1.01	<0.001
Attitude score	0.81	0.58	0.66 to 0.97	<0.001

**Table 3 healthcare-13-02257-t003:** Multivariate linear regression for factors of perceived disaster preparedness.

Variable	Coefficient (B)	Standardized Coefficient (Beta)	95% CI for B	*p*
Gender	1.94	0.1	0.34 to 3.54	0.018
Years of experience	−0.12	−0.08	−0.25 to 0.02	0.08
Profession	−0.71	−0.09	−1.37 to −0.05	0.04
Trauma courses	−0.06	−0.003	−1.91 to 1.80	0.95
Any Disaster training	−0.28	−0.04	−0.95 to 0.40	0.42
In-department training	−0.31	−0.01	−2.57 to 1.96	0.79
Participation in education	0.20	0.04	−0.39 to 0.79	0.52
Previous participation in disaster management	2.60	0.14	1.005 to 4.20	0.002
Between-institution collaborations	4.49	0.10	0.68 to 8.30	0.02
A disaster plan set in place at the institutional level	4.00	0.12	1.23 to 6.78	0.005
Knowledge score	0.63	0.53	0.51 to 0.76	<0.001
Attitude score	0.43	0.31	0.30 to 0.56	<0.001

## Data Availability

The data that support the findings of this study are available from the corresponding author upon reasonable request.

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
