# Peer review of "Preparedness for Disaster Response: An Assessment of Northeast Romanian Emergency Healthcare Workers"

_healthcare, 2025, doi:10.3390/healthcare13182257_

Round 1
Reviewer 1 Report
Comments and Suggestions for Authors
Dear Author. Thank you so much for conducting such an important study. It is indeed an excellent addition to the literature. However, some modifications needed
- The introduction is very lengthy and irrelevant to the topic sometimes. Kindly rewrite and make it more coherent. As I can see there are abrupt jumps from one topic to other and then there is repetition of fact in different paragraphs
- Although the English is pretty well but there ae some typos such as liker instead of Likert.
- Methodology section is unclear – how you get emails? How many hospitals were targeted and where you get emails or what kind of WhatsApp groups? The participation was voluntary or what ?
- Kindly mention the study site and selection criteria
- Results were over described as they are already present in Table 1 so don’t mention the same in paras
- Somewhere you wrote K and some places you wrote Knowledge
- You mentioned total 212 in methodology whereas in table 1 total number was 211
- Disaster management education received- have you defined each education option?
- The most like disaster to happen-cyberterrorism? What does a HCW have to do with cyberterrorism?
- This is in line with other recently reported results in developed countries like the UAE (United Arab Emirates) and in developing countries like Pakistan (Gillani et al., 2021; Shanableh et al.,2023)- give reference for other developed countries like China and developing countries like Yemen.
- I my opinion the discussion is very ling with many unnecessary references and facts. Kindly concise it according to the main findings and compare them with previous not all results
Author Response
Kindly see the attachment.

Reviewer 2 Report
Comments and Suggestions for Authors
Thank you for your hard work on an interesting paper.
A few points
Some disaster are predictable - Hurricane Katrina for example
Experts consider that there is no such thing as a natural disaster - they are all man-made - we live and build in unsafe places, we ignore environmental factors and we marginalise large groups in our communities https://www.google.com/url?sa=t&source=web&rct=j&opi=89978449&url=https://www.undrr.org/our-impact/campaigns/no-natural-disasters&ved=2ahUKEwiG__Plv4mPAxUbrVYBHcHCH3gQFnoECDEQAQ&usg=AOvVaw0VwkQQ_4y3kh82h572h1c9
The methodology is clear thank you. As was the results.
Line 174 Declaration of Helsinki needs a date please
In the table time since the last disaster course Never – does that mean they have not had any disaster training? Please clarify
Disaster training organised by the dept might sit better in the table above time since last disaster course
Line 291 – do you have a reference for this statement?
I wonder if cost of the courses on Disaster management has any impact on the attendance at one of these courses?
Thank you again for your hard work – it is appreciated!
Author Response
Kindly see the attachment

Reviewer 3 Report
Comments and Suggestions for Authors
Dear authors
The manuscript presents a study on disaster response and preparedness, a critical area of research, particularly in the context of emergencies. The significance of this topic cannot be overstated, as effective crisis management can substantially impact outcomes during health emergencies.
Comments and Recommendations:
1. Diversity of Respondents:
The survey conducted in this study included responses from a doctor, a nurse, and a paramedic. Each of these professional roles possesses distinct responsibilities, expertise, and perspectives regarding crisis response and preparedness. It is recommended that the authors consider conducting a subgroup analysis to explore potential differences in responses among these professionals. This would not only enhance the depth of the analysis but also provide valuable insights into the varying perceptions and experiences of different healthcare providers in emergency situations.
2. Methodological Considerations:
While the current methodological framework is commendable, further elaboration on the sampling strategy and the rationale for selecting the specific respondents would strengthen the study. Clarifying how these individuals were chosen could provide context regarding the representativeness of the sample and the generalizability of the findings.
3. Data Analysis:
The analytical techniques employed in the study should be explicitly detailed. In particular, it would be beneficial to outline any statistical methods used for subgroup analysis if conducted. This transparency will enhance the rigor of the research and allow for reproducibility.
4. Literature Contextualization:
The manuscript would benefit from a more comprehensive review of existing literature on crisis response and preparedness. Integrating relevant studies could provide a stronger theoretical framework for the research and highlight how this study contributes to the existing body of knowledge.
5. Implications for Practice:
The discussion section should more thoroughly address the practical implications of the findings. Given the critical nature of crisis response, outlining specific recommendations for training or policy adjustments based on the results could significantly enhance the manuscript's impact.
6. Conclusion:
The conclusion should succinctly summarize the key findings while also emphasizing their relevance to future research and practice in crisis management. It may also be beneficial to suggest avenues for further investigation based on the limitations identified in the study.
I look forward to seeing the revised version.

Reviewer 4 Report
Comments and Suggestions for Authors
Dear Authors,
I would love to know the questions you used to assess the knowledge and attitudes of your study participants. This should be included as an appendix to the article. Without this I cannot fully comment on your results.
I'm totally agree with you that preparing for the unexpected allows for better execution of the most difficult tasks in practice. I believe this article will help Romania improve its disaster preparedness.
Author Response
Kindly see the attachment.

Reviewer 5 Report
Comments and Suggestions for Authors
Dear authors,
I had the privilege to read your paper titled "Readiness for Disaster Response: An Assessment of Northeast Romanian Emergency Healthcare Workers", aiming at shedding light on some key indicators of performance of healthcare workers potentially involved in disasters in a Romanian district.
Hereafter some suggestions to improve it:
Overall: well written and balanced.
Please double check acronyms since sometimes a word is skipped (e.g., line 71 in introduction healthcare workers -> HCW, and elsewhere).
Also double check the way "Iasi" county is written differently along the text.
Abstract:
- intro: the second sentence implies something from the first sentence but to be clarified to the reader, and it could be ambiguous; so that preparedness is achieved through training. Make it more explicit please, to avoid any possible misinterpretation by the reader.
Keywords: better to use "Disaster Preparedness" –more correlated with the MeSH term "Disaster Planning" than the current "Disaster Readiness".
Introduction
- suggest using a unique key-term for "preparedness" across the text. E.g. at line 51 do not use "readiness" but "preparedness". There is also a bit of confusion when describing HCW "readiness" and "preparedness" because these two concepts can be slightly different. If the authors instead want to use both, they should define them for clarity of the reader.
- Line 78: Please do not cite Wikipedia as a reference. Despite the extremely high consideration I have for Wikipedia, cite the direct source for data–if not from an academic peer-reviewed source, please cite governmental or EU gov' sources.
- Line 78: try to smoothen the passage since it's a slight hard to connect the high hazard of Romania and the improvement in emergency care this way. Maybe a suggestion (in bold) "The em care has therefore seen consistent..."?
- Third paragraph: maybe worth mentioning also the WHO EMT initiatives and the EU Civil Protection?
Methods
- line 112: are -> were
- line 116: properly cite software (see journal's or publishing guidelines)
- line 125: verb is missing from this sentence; also, did the authors take into account the role (doctor, paramedic, nurse, etc?)
- Likert scales and description of answers interpretation: please state first something like "A five-point Likert scale was used to assess the three domains" and then attach the already present definitions of the points 1 to 5. Then try to synthesize interpretation according to the 3 different numbers of answers. Do not use numbers to start new sentences (e.g., 13 -> Thirteen).
Results
- line 197: have-> had; number 134 beginning sentence, please turn into a word
- line 224: double check commas to be replaced with dots in the IQRs
Table 1
- "trauma life support courses" and last row of the table: please replace comma with a dot
- "The most likely disaster scenario to happen" -> percentages don't add up please double check (final: 99,535 %)
Discussion, Limitations, Conclusions
- line 260: please moderate the sentence by adding "included in the present study" when referring to the sample
- Same minor issue should be reported in limitations – since the participants were recruited from the Iasi County, and similarly moderating the statement at line 385 as already suggested for line 260.
References: remember to put references in numbers sequentially in the text, and revise the reference section at the end as per the journal's recommendations.
Hope you can revise the paper accordingly, eager to read again the revised copy.
Kind regards.
Author Response
Kindly see the attachment.

Round 2
Reviewer 4 Report
Comments and Suggestions for Authors
Dear Authors,
Congratulations on the work well done on the text corrections. The article has benefited greatly from these changes.